# Prevalence and Treatment of Vitamin D Deficiency in Young Male Russian Soccer Players in Winter

**DOI:** 10.3390/nu11102405

**Published:** 2019-10-08

**Authors:** Eduard Bezuglov, Aleksandra Tikhonova, Anastasiya Zueva, Vladimir Khaitin, Zbigniew Waśkiewicz, Dagmara Gerasimuk, Aleksandra Żebrowska, Thomas Rosemann, Pantelis Nikolaidis, Beat Knechtle

**Affiliations:** 1Department of Sport Medicine and Medical Rehabilitation, Sechenov First Moscow State Medical University (Sechenov University), 119435 Moscow, Russia; 2FC Lokomotiv, 107553 Moscow, Russia; 3Russi An Football Union, 115172 Moscow, Russia; 4FC Zenit, 197341 Saint-Petersburg, Russia; khaitinvladimir@gmail.com; 5Institute of Sport Science, Jerzy Kukuczka Academy of Physical Education, 40-065 Katowice, Poland; z.waskiewicz@awf.katowice.pl; 6Department of Sports Training, Jerzy Kukuczka Academy of Physical Education, 40-065 Katowice, Poland; d.gerasimuk@awf.katowice.pl; 7Department of Physiological and Medical Sciences, Jerzy Kukuczka Academy of Physical Education, 40-065 Katowice, Poland; a.zebrowska@awf.katowice.pl; 8Institute of Primary Care, University of Zurich, 8091 Zurich, Switzerland; thomas.rosemann@usz.ch; 9Exercise Physiology Laboratory, 18450 Nikaia, Greece; pademil@hotmail.com; 10Medbase St. Gallen Am Vadianplatz, 9001 St. Gallen, Switzerland; beat.knechtle@hispeed.ch

**Keywords:** vitamin D_3_, cholecalciferol, vitamin D deficiency, treatment of vitamin D_3_ deficiency, soccer

## Abstract

Vitamin D (25(OH)D) insufficiency and deficiency are highly prevalent in adult soccer players and can exceed 80% even in regions with high insolation; however, the treatment of this condition is often complicated. The aim of the present study was to examine the prevalence of vitamin D insufficiency and deficiency in youth Russian soccer players and the efficacy of its treatment. Participants were 131 young male football players (age 15.6 ± 2.4 years). Low vitamin D levels (below 30 ng/mL) were observed in 42.8% of the analyzed participants. These athletes were split in two groups composed of persons with vitamin D deficiency (serum vitamin D below 21 ng/mL) and insufficiency (serum vitamin D in range of 21–29 ng/mL). A dietary supplement of 5000 IU cholecalciferol per day was administered for two months. After the treatment, an average 92% increase in vitamin D concentration was observed (before treatment—19.7 ± 5.4 ng/mL, after treatment—34.7 ± 8.6 ng/mL, *p* < 0.001) and 74% of the post-treatment values were within the reference range (30–60 ng/mL). Serum concentration of vitamin D increased by 200% ± 98% (*p* < 0.001) during the first month of treatment with vitamin D deficiency and insufficiency being successfully treated in 83% of the football players. In summary, the prevalence of vitamin D insufficiency and deficiency was high in young Russian soccer players. Furthermore, it was indicated that the daily usage of cholecalciferol in a dose 5000 IU was an effective and well-tolerated treatment for vitamin D insufficiency. No linear dependency between the duration of treatment and increase in vitamin 25(OH)D concentration was observed.

## 1. Introduction

Vitamin D plays a crucial role in phosphorus and calcium metabolism and thus affects the state of bone tissue. It is important for the function of the immune system and helps maintaining blood glucose levels, structure of connective tissues and muscle tonus [1,2]. The multirole nature of vitamin D action is explained by the location of its receptors, which can be found in multiple tissues [3]. Chronic vitamin D deficiency in childhood can lead to the development of rickets or osteomalacia. With less severe insufficiency, processes leading to bone resorption and development of osteoporosis are reinforced, which may lead to an increase in frequency of fractures [4]. Most of the studies on vitamin D status in different population groups show that its deficiency directly affects strength and muscle mass as well as muscle trauma rates [5,6]. Adequate vitamin D (25(OH)D) levels (i.e., 40 ng/mL and above) are essential for the prevention of bone injury, including stress fractures [7].

Most of the vitamin D found in the human body is synthesized when UV rays penetrate the open skin at a specific angle. The skin may provide up to 80–100% of the required vitamin D. At the same time, it is very difficult to ensure an adequate intake of this vitamin only through food as its dietary content is quite low [5]. The leading risk factors for vitamin D deficiency are dark skin color, insufficient insolation, obesity, malabsorption syndrome and old age [8,9,10,11,12]. The most accurate method for determining the biochemical markers of vitamin D can be considered isotope dilution liquid chromatography mass spectrometry, however, current immunoassays demonstrated acceptable performance [13]. Despite the lack of proper standardization, a serum concentration of 25(OH)D is most often used to assess the status of vitamin D in the human body, although in recent years a number of other biochemical agents (vitamin D-binding protein, free/bioavailable 25(OH)D and parathyroid hormone et al.) have been proposed for this purpose [14].

The prevalence of vitamin D deficiency varies by region of residence and depends on the amount and duration of exposure to UV rays [15]. It is common for the regions located to the north of the 35th parallel north, e.g., most of the territory of Russia and Finland. In the research of the Lehtonen-Veromaa et al. with taking part in it 131 gymnasts and runners, who reside at latitude 60 degrees north, the deficiency of the serum concentration of the biochemical marker of vitamin D, 25(OH)D, was found in more than 80% of the participants [16]. This phenomenon occurs because the angle at which sun rays enter the atmosphere becomes more shallow, which leads to their dissipation [17]. Vitamin D begins to exert its effects when its level exceeds 30 ng/mL [18]. However, according to the research of the Butscheidt et al., for the effects associated with muscle tissue to develop, its concentration has to exceed 40 ng/mL, maximum impact on physical performance is realized at the concentration of at least 50 ng/mL [18]. 

However, there are evidences that the positive effects of vitamin D on muscles and immune function are possible even at a concentration of 25(OH)D of about 20 ng/mL [19]. The vitamin then begins to be stored in muscles for its later utilization [20]. There are two forms of vitamin D, ergocalciferol (vitamin D_2_) and cholecalciferol (vitamin D_3_). Vitamin D_3_ (vitamin 25(OH)D) is an important agent used in clinical setting to prevent and treat vitamin deficiencies [21].

There are studies confirming that vitamin D_3_ more efficient (by 87%) for the correction of reduced serum concentrations of 25(OH)D comparing with vitamin D_2_ [22]. Vitamin D deficiency is very common among professional athletes, where it reaches 60% to 80%. Such a deficiency may negatively impact their performance and increase injury rate [23,24,25,26,27]. An insufficient concentration of 25(OH)D is often found among recreational athletes, reaching 76% in this group [28]. The existing studies performed in adult professional soccer players also confirmed the high prevalence of vitamin D insufficiency and deficiency in this group of athletes. This condition was often diagnosed even in the regions with high insolation [15,29]. It has been shown that increasing the amount of insolation is more effective than introducing vitamin D dietary supplements as a treatment [30]. In the general population, various therapeutic schemes for vitamin 25(OH)D deficiency are used, differing in duration, method of administration and starting dose. According to various experts, daily doses during initial therapy for vitamin D insufficiency and deficiency of various levels can fluctuate from 2000 to 200,000 IU daily. At the same time, 10,000 IU is considered to be the upper limit of daily intake of vitamin D. Most commonly, a dose of 50,000 IU vitamin D weekly is used for the rapid treatment of its deficiency [31].

A number of studies exists where athletes were prescribed weekly doses of 12,500–40,000 IU of vitamin D in different modes over a period of 8 to 12 weeks. In these studies, a 66–270% elevation of plasma vitamin D concentration was achieved. Notably, it was more effective to administer 5000 IU vitamin D daily than 40,000 IU vitamin D once a week [23,24,32]. High physical activity specific to athletes can increase the physiological need for vitamin D. Typically, an human body requires about 3000–5000 IU vitamin D daily under the usual load [31].

Currently, no data exist on vitamin D deficiency among young healthy Russian soccer players who reside at latitude 55 degrees north and are therefore at a high risk of developing this condition. At the same time, it should be noted light days are short and temperatures are low during winters in Russia. Most of the training takes place indoors, further increasing the probability of vitamin D deficiency development. We assume that in this large category of athletes, who residing in Russia, insufficiency and deficiency of vitamin D in winter will be common. That is why an analysis of the prevalence of this condition, as well as a search for safe and well-tolerated treatment schemes utilizing the most available oral vitamin D supplements, is therefore of high practical importance.

## 2. Materials and Methods

### 2.1. Ethical Approval

The protocol of the study was approved by the official Local Ethics Committee of the Sechenov First Moscow State University with the number 11–19 of 25 July 2019.

### 2.2. Subjects

This study summarizes the data obtained in a cohort of 131 white male soccer players from Football School Lokomotiv and FC Lokomotiv Moscow Youth team aged 12 to 23 years (mean age 15.6 ± 2.4 years) who had no contraindications for sports. The study included young soccer players resided in Moscow, a city located at latitude 55 degrees north. All the athletes have been playing football since the age of 6–7 and for at least 5 years before the research, regularly (6–7 times a week, 11 months a year) train 90–120 min a day and participate in matches of their teams. All participants of the study provided their informed consent. Consent from the parents of all study participants under 18 years of age was obtained. Athletes who were 18 years or older provided the consent form directly. All stages of the study comply with the legislation of the Russian Federation. The study was conducted in December 2018 to February 2019 at the Lokomed Medical Center of Lokomotiv FC Moscow with the participation of the staff of the Department of Sports Medicine and Medical Rehabilitation of the Sechenov First Moscow State Medical University.

### 2.3. Criteria for Exclusion from the Study

The criteria for exclusion from the study were as follows: the athlete received vitamin D supplements 30 days or less prior to first blood sampling, the athlete received other supplements during the study, the athlete suffered from acute respiratory viral infections or any other diseases that resulted in absence from three or more training sessions 30 days or less prior to the examination, the athlete could not maintain daily contact with the medical personnel distributing vitamin D_3_ supplements, the athlete spent more than three days outside Moscow during the last three months, and the athlete was expelled from the academy during the study.

### 2.4. Laboratory Testing

Three blood samples were obtained. The first sample was obtained in December 2018, and the following samples were obtained on the 35th and 70th days of the study (following a 5-day pause after the first and the second month of therapy). Therefore, after 30 days of cholecalciferol therapy, the participants of both groups did not receive it for five days.

Fasting blood samples were collected from the cubital vein in the morning. Three immunoassay blood tests for vitamin D_3_ (25(OH)D) were conducted using an in vitro reagent set for 25(OH)D produced by Euroimmun AG (Germany) and a Mindray MR-96A microplate reader (China). According to the actual guidelines, diagnostic criteria for vitamin D deficiency and insufficiency were serum vitamin D below 21 ng/mL and in range of 21–29 ng/mL, respectively [33]. Values of 30–60 ng/mL were considered as normal. The serum concentration of vitamin D more than 60 ng/mL was considered as simply higher than normal indicators.

Body height and body weight measurements were obtained from all participants. In the groups with deficient, insufficient and excessive vitamin D, muscle and body fat mass measurements were obtained using bioimpedance analysis on the day following the first blood sampling procedures. The ABC-02 “MEDASS” (Russia) analyzer was used for bioimpedance analysis. The procedure was performed in the morning before the treatment, with the patient in the fasting state.

### 2.5. Supplementation with Vitamin D

Based on the results of the 25(OH)D blood test, athletes were assigned to one of two groups (vitamin D deficiency, group 1; vitamin D insufficiency, group 2), and treatment started. Athletes with the serum concentration of 25(OH)D higher than 30 ng/mL did not take part in the further research, because the main purpose of this research was to study the prevalence of vitamin D deficiency and insufficiency among young football players and then to evaluate the effectiveness of correction of the low content of 25(OH)D with daily supplementation of oral cholecalciferol. Vitamin D deficiency and insufficiency was treated with 5000 IU oral cholecalciferol (SiS Vitamin D_3_, 5000 IU, United Kingdom) daily after breakfast. Treatment lasted for 60 days, with a 5-day break after the 30th day of treatment and was supervised by the medical staff of the Football School daily.

### 2.6. Statistical Analysis

IBM SPSS Statistics software v.23.0 (IBM, Armonk, NY, USA) was used for statistical analysis. Normality of the collected fata was tested using the Kolmogorov-Smirnov test. A two-sample independent T-test was used to assess the intergroup difference in case of normal distribution (body mass, height, BMI, BF muscle mass for group 1 and 2, body mass for groups of athletes with serum concentration of 25(OH)D below 30 ng/mL and above 60 ng/mL had characteristics of normal distribution). Mann-Whitney U-test was used to assess significance of intergroup difference for non-normal distribution (height, BMI, BF muscle mass for groups of athletes with serum concentration of 25(OH)D below 30 ng/mL and above 60 ng/mL). To compare values of serum concentration of 25(OH)D before treatment and after 30 days and 60 days of the treatment in group 1, group 2 and in both groups, one-way ANOVA method was used followed by the Tukey test result. Values at *p* < 0.05 were considered statistically significant.

## 3. Results

Vitamin D insufficiency and deficiency was highly prevalent in the analyzed population, with a reduced vitamin D plasma concentration observed in 42.8% of participants (56 soccer players). Vitamin D insufficiency was found in 19.9% (26) of the participants, and vitamin D deficiency in 22.9% (30) of the participants. Vitamin D level was within normal range in 26.7% (35) of the young soccer players, while in 30.5% (40) of the players it reached 61–130 ng/mL and was above reference values (Figure 1).

We compared weight, height, body mass index (BMI), body fat mass and skeletal muscle mass among athletes with serum concentration of 25(OH)D below 30 ng/mL and above 60 ng/mL. There was no statistically significant difference (Table 1).

Athletes with lower-than-normal levels of serum vitamin 25(OH)D were split in two groups composed of persons with vitamin D deficiency (group 1) and insufficiency (group 2). Two and five players were excluded from groups 1 and 2, respectively, during the first five days of the study, as these individuals were expelled from the academy and were unavailable for sampling. The second and third blood samples were obtained 35 and 70 days after the beginning of therapy, respectively. Seven and three of the 49 participants, respectively, were unavailable for the second and third sampling. Their absence was caused by their studies. All these athletes fully adhered to the therapeutic program. Group 1 consisted of 24 individuals (mean age 15.2 ± 2.4 years) with a mean vitamin 25(OH)D level of 15.2 ng/mL. Group 2 consisted of 25 people (mean age 14.5 ± 1.5 years) with a mean vitamin 25(OH)D level of 23.9 ng/mL. The mean vitamin level for both groups combined was 19.6 ng/mL. There was no statistically significant difference in weight, height, body mass index (BMI), body fat mass and skeletal muscle mass between the two groups (Table 2).

An average 92% increase in mean vitamin 25(OH)D level for both groups combined was observed after 60 days of treatment with cholecalciferol (19.7 ± 5.4 ng/mL to 34.7 ± 8.6 ng/mL, *p* < 0.001). Mean serum 25(OH)D increased by an average 138% reaching 34.8 ± 9.6 ng/mL (*p* < 0.001) in Group 1 and by an average 44% reaching 34.5 ± 7.7 ng/mL (*p* = 0.002) in Group 2. Vitamin D level was normalized in 34 (74%) of soccer players from both groups. Normal values were achieved in 70% (16) members of group 1 and in 78% (18) members of group 2. In 12 people (26%) from both groups combined serum level of vitamin D_3_ remained below the reference values despite the observed increase. Notably, the effect of therapy was most pronounced after 30 days of cholecalciferol administration, when mean vitamin 25(OH)D level in both groups combined increased by an average 134% (from 19.7 ± 5.4 ng/mL to 42.4 ± 13.2 ng/mL, *p* < 0.001). Vitamin D serum concentration increased by an average 200% (from 15.3 ng/mL to 41.7 ng/mL, *p* < 0.001) in group 1 and by an average 80%% (from 24 ng/mL to 43 ng/mL, *p* < 0.001) in group 2. Subsequent administration of vitamin D_3_ did not lead to an expected increase in mean concentration. On the contrary, it dropped by an average 8.7% (from 42.4 ng/mL to 34.7 ng/mL, *p* = 0.014) in group 1. Figure 2 shows dynamic of serum concentrations of vitamin 25(OH)D in the groups. No gastrointestinal side effects or allergic reactions were observed in any of the soccer players during therapy. 

## 4. Discussion

The study showed that reduction in vitamin D serum concentration is highly prevalent in young soccer players residing at latitude 55 degrees north. Also, in the study we made a correction with the same doses of cholecalciferol both in soccer players with deficiency and with insufficient serum concentration of 25(OH)D and the maximum effectiveness of therapy was shown in the group of soccer players with a lower concentration of 25(OH)D. At the same time, the effectiveness of the correction was uneven for two months and decreased over time. Moreover, was identified a group of athletes in whom the level of 25(OH)D did not increase against the background of correction.

The findings are consistent with other studies that also revealed a widespread prevalence of 25(OH)D deficiency and insufficiency among athletes. Hamilton et al. [34] studied a cohort of 342 adult soccer players from the Middle East. Vitamin D insufficiency was observed in 90% of the participants. The authors explained such a high prevalence by the fact that, although the athletes lived in the region with high insolation, they only spent about 30 min outdoors [34]. Other possible reasons included exercising in the evening, use of sunscreen and wearing clothes that cover most of the body. Krzywanski et al. [30] performed an analysis of vitamin D status in 409 Polish athletes who trained both indoors and outdoors. During winter season, vitamin D insufficiency was observed in more than 80% of the athletes. At the same time, in this group of athletes, solar insolation increased the level of 25(OH)D by 85%, and taking supplements containing vitamin D_3_ only by 45%. In summer, the prevalence of insufficiency significantly lowered in athletes who trained outdoors, while in athletes who trained indoors it remained high [30].

The high prevalence of low levels of vitamin D was also found in the studies on Swedish young soccer players and Canadian young hockey players residing in regions with low levels of solar insolation [34,35]. However, prolonged insolation is not a guarantee of sufficient formation of vitamin D and its deficiency is also widespread among athletes permanently residing in the southern regions [36]. Reduced workload in comparison to summer and autumn may provide a possible explanation of lower prevalence of vitamin D deficiency among young Russian soccer players during winter season. Additionally, all participants of this study were supervised by coaches and doctors, making timely correction of workloads possible. Excessive load may be one of the factors causing plasma vitamin D reduction [31]. Lately, multiple therapeutic schemes for vitamin D insufficiency have been proposed for various population groups, which utilize both injectable and oral forms. A choice of scheme aimed at normalizing 25(OH)D serum level should be based on safety, tolerability and time required to achieve the necessary vitamin concentration. 

According to the available data, the concentration for vitamin D should exceed 40 ng/mL. At this concentration, the vitamin exerts a beneficial effect on muscle tissue and reduces the frequency of stress damage to bone tissue [7,24]. Therefore, 40 ng/mL should be selected as a minimum target level during therapy. Relatively high drug doses during both initial therapy and maintenance therapy are a characteristic feature of treatment schemes for vitamin D deficiency in athletes. A systematic review by Farrokhyar et al. [37] analyzed 13 randomized controlled trials where effectiveness of various treatment schemes of vitamin D insufficiency in athletes was studied. It has shown that, during winter season, a positive result may be achieved in athletes residing to the north of the 45th parallel by administering 5000 IU vitamin D_3_ daily. This therapeutic scheme resulted in an increase in serum 25(OH)D of 27.8 ng/mL, while administering 3000 IU daily only led to a 15.2 ng/mL increase. A total of 532 of the athletes (311 in the vitamin D group and 221 in the placebo group) met all the required inclusion criteria for the study [37].

Skalska et al. found a marked increase in the serum concentration of 25(OH)D with a daily intake of about 5000 IU for 8 weeks in their study on young football players. However, no side effects were observed [38]. Teixeira et al. [32] used 25,000 IU cholecalciferol once in two weeks for eight weeks to treat a group of 28 soccer players with reduced plasma vitamin D. At the end of treatment, they achieved a 70% increase in serum concentration of 25(OH)D compared to the pre-treatment state [32]. However, the use of high doses (70,000 IU per week for 3 months) may have an adverse effect due to the inhibition of the biological activity of 1,25(OH)_2_D_3_ [39].

In most of the existing studies, the therapy lasted for 8 to 12 weeks, and serum vitamin levels were analyzed at the beginning and end of the study. In this study, three blood samples were obtained. A non-linear nature of changes in vitamin D concentration was revealed, with the peak occurring on the thirtieth day of therapy. The therapy was found to be more effective in a group composed of athletes with vitamin 25(OH)D deficiency, where a 138% increase in vitamin D concentration was achieved. This exceeds the results shown in the most of previously published studies. At the same time, the achieved vitamin D levels were still lower than the minimum of 40 ng/mL required for vitamin D to beneficially influence both muscle and bone tissue [40].

The limitations of this study include the lack of control over changes in the serum concentration of 25(OH)D in the group of soccer players with normal and elevated levels. Only one dosage of cholecalciferol was used for correction of deficiency and insufficiency of 25(OH)D, which makes it impossible to identify the minimum possible effective doses of supplements containing vitamin D_3_. Future works should be aimed at studying changes in the serum concentration of 25(OH)D in this group of athletes during a long period of time, as well as determining the optimal doses of cholecalciferol for successful correction of the low content of 25(OH)D.

## 5. Conclusions

In this study high prevalence of deficiency and insufficiency of 25(OH)D in winter among young soccer players permanently residing at 55 degrees north latitude was revealed. It was established that the efficiency of the correction of these conditions depends on the initial serum concentration of 25(OH)D and varies during treatment. Further research should be aimed at optimizing the existing treatment schemes for vitamin 25(OH)D deficiency and developing guidelines for the elimination of possible risk factors.

## Figures and Tables

**Figure 1 nutrients-11-02405-f001:**
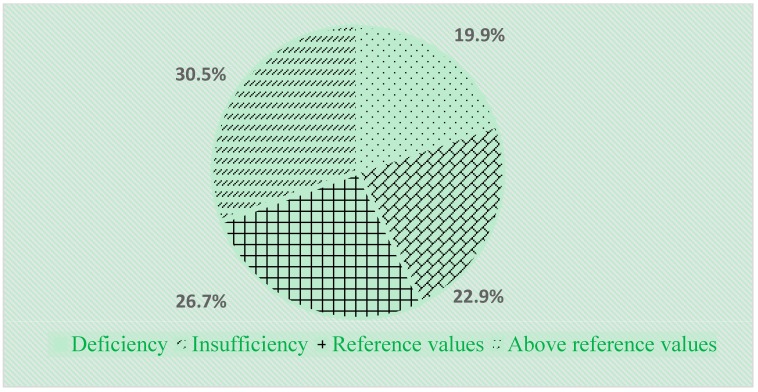
Serum level of vitamin 25(OH)D in young soccer players permanently residing in Moscow grouped by status.

**Figure 2 nutrients-11-02405-f002:**
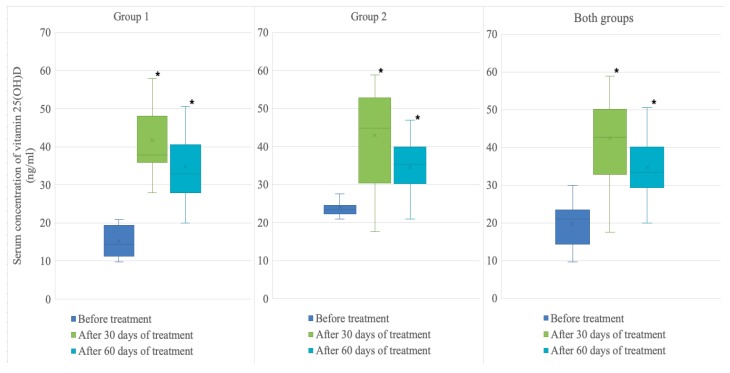
Serum concentration of vitamin 25(OH)D (ng/mL) before treatment and after 30 and 60 days of treatment in Group 1, Group 2 and both groups combined.

**Table 1 nutrients-11-02405-t001:** Anthropometric characteristics and body composition in groups of athletes with serum concentration of 25(OH)D below 30 ng/mL and above 60 ng/mL.

Parameter	Below 30 ng/mL	Higher than 60 ng/mL	*p*-Value
Body mass, kg	64.6 ± 11.8	65.0 ± 16.0	0.88
Height, cm	175.0	175.7	0.83
Body mass index (BMI), kg·m^−2^	20.9	21.3	0.17
Body Fat Mass, %	16.3	16.2	0.88
Muscle mass, %	56.6	56.9	0.68

**Table 2 nutrients-11-02405-t002:** Anthropometric characteristics and body composition in Group 1 and 2.

Parameter	Group 1	Group 2	*p* Value
Body mass, kg	67 ± 10	62 ± 13	0.50
Height, cm	177 ± 9	173 ± 12	0.85
BMI, kg·m^−2^	21 ± 2	20 ± 2	0.57
Body Fat Mass, %	16 ± 4	16 ± 4	0.27
Muscle mass, %	57 ± 2	56 ± 8	0.17

Values are presented as mean ± SD; BMI = body mass index; BF = body fat percentage.

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
