# Peer review of "Prevalence and Treatment of Vitamin D Deficiency in Young Male Russian Soccer Players in Winter"

_nutrients, 2019, doi:10.3390/nu11102405_

Round 1
Reviewer 1 Report
Overall Comments:
The authors should be commended for undertaking this study on a large group of young athletes. Vitamin D is important for physical development, particularly skeletal health.
I do have some reservations about this study however which I am hoping the authors can clarify and rectify.
Individual comments:
Line 27: why is this “often complicated”
Lines 32- 37: please report values with standard deviation and significance
Line37: I am not sure that you can say “high”, when <20% of the participants were deficient
Line 38: 5,000 IU is not a “small dose”
Line 66: “when its level exceeds 30 ng/ml” please provide a reference.
Lines 74-76: please provide references to support your statement that exercises increase the risk of developing vitamin D deficiency.
lines 84-88: the authors present evidence to support their use of 5,000 IU of vitamin D3 in their intervention. They have presented old evidence based upon the use of vitamin D2, not D3 and as such have provided a high daily dose of vitamin D3 to their participants (Holick, M.F. The vitamin d epidemic and its health consequences. Journal of Nutrition 2005, 135, 330 2739S-2748S)
Vitamin D3 has an estimated potency up to 4 times higher than vitamin D2. I refer the authors to: Heaney, R. P., Recker, R. R., Grote, J., Horst, R. L., & Armas, L. A. G. (2011). Vitamin D3 Is More Potent Than Vitamin D2 in Humans. The Journal of Clinical Endocrinology & Metabolism, 96(3), E447-E452. doi:10.1210/jc.2010-2230 (https://academic.oup.com/jcem/article/96/3/E447/2597204).
line 93-94: Further the authors state that athletes require 3,000 to 5,000 IU of vitamin D and cite Holick, M.F. The vitamin d epidemic and its health consequences. Journal of Nutrition 2005, 135, 330 2739S-2748S – This must be an incorrect reference as that paper does not discuss athletes.
Lines 102-103: Please clarify your hypothesis.
Lines 104-144: The methodology for the study is unclear and needs to be better described.
It appears that the actual design of the study was as follows: Control group – those with vitamin D serum levels over 30 ng/ml Supplemented group – those with serum vitamin D levels below 30 ng/mlIf you had supplemented the insufficient and deficient groups with different doses of vitamin D, they would have represented different groups.
It also appears that the Authors have taken blood samples from 131 participants and then excluded over 50% of the participants based upon their vitamin D sufficiency levels. This has not been clearly stated in the methodology or in the exclusion criteria.Can you please explain if this is actually what happened?
Please explain why you selectively measured muscle and body fat measurements for insufficient and deficient players only. Please explain why you chose to implement a five day break in the middle of the supplementation period. Please explain whether all participants had blood taken at every time point, and if not, explain why.Lines 145 – 193: The authors have selectively reported the results of those participants that received a supplement and have not presented the results of those participants who presented with normal levels at baseline with the exception of reporting the baseline vitamin D status.
It is unclear whether those participants who were deemed to be sufficient were included in the study beyond baseline.
It would have been informative to see what the progression of vitamin D levels were for all participants over the course of the intervention.
You should include ALL results in your paper and provide explanation in the Discussion section.
Figure 1 – please change to patterned areas not colour as the sections of the pie chart cannot be distinguished if printed in black and white.
Lines 194 to 252:
The discussion has not compared the study’s results well to other literature.
Please provide discussion to explain what mechanisms may have caused the results in the second month to be lower than the first month.
Please provide discussion regarding limitations to the study.
Line 200: please explain what “other biochemical agents” you are refering to.
Line 203: references for “other studies”
Line 226: “According to the available data” – please provide a reference and explain why you used a cut-off point of 30 ng/ml if the concentration should exceed 40 ng/ml.
Conclusion
I am unsure how the findings support the conclusion. Please clarify.
Author Response
The authors should be commended for undertaking this study on a large group of young athletes. Vitamin D is important for physical development, particularly skeletal health.
I do have some reservations about this study however which I am hoping the authors can clarify and rectify.
Individual comments:
Line 27: why is this “often complicated”
Answer: Due to lack of the safe, well-tolerated treatment of this condition and gold standard which were put into the regular and daily practice among sportsmen.
Lines 32- 37: please report values with standard deviation and significance
Answer: We agree with the expert reviewer and changed as requested.
Line37: I am not sure that you can say “high”, when <20% of the participants were deficient
Answer: We agree with the expert reviewer. We meant the prevalence of both, deficiency and insufficiency, which in total amount are more than 40 %. We changed as requested
Line 38: 5,000 IU is not a “small dose”
Answer: We agree with the expert reviewer. We deleted the word «small».
Line 66: “when its level exceeds 30 ng/ml” please provide a reference.
Answer: We agree with the expert reviewer. We added the reference in the text.
Lines 74-76: please provide references to support your statement that exercises increase the risk of developing vitamin D deficiency.
Answer: We agree with the expert reviewer. It was our assumption, because during hard physical work, training, human body expend more energetic resources, vitamins than during daily activities. Thus, the athlete’s organism needs more vitamins, including vitamin D.
lines 84-88: the authors present evidence to support their use of 5,000 IU of vitamin D3 in their intervention. They have presented old evidence based upon the use of vitamin D2, not D3 and as such have provided a high daily dose of vitamin D3 to their participants (Holick, M.F. The vitamin d epidemic and its health consequences. Journal of Nutrition 2005, 135, 330 2739S-2748S)
Vitamin D3 has an estimated potency up to 4 times higher than vitamin D2. I refer the authors to: Heaney, R. P., Recker, R. R., Grote, J., Horst, R. L., & Armas, L. A. G. (2011). Vitamin D3 Is More Potent Than Vitamin D2 in Humans. The Journal of Clinical Endocrinology & Metabolism, 96(3), E447-E452. doi:10.1210/jc.2010-2230 (https://academic.oup.com/jcem/article/96/3/E447/2597204).
Answer: We agree with the expert reviewer and changed as requested.
line 93-94: Further the authors state that athletes require 3,000 to 5,000 IU of vitamin D and cite Holick, M.F. The vitamin d epidemic and its health consequences. Journal of Nutrition 2005, 135, 330 2739S-2748S – This must be an incorrect reference as that paper does not discuss athletes.
Answer: We agree with the expert reviewer and changed as requested.
Lines 102-103: Please clarify your hypothesis.
Answer: We agree with the expert reviewer and changed as requested.
Lines 104-144: The methodology for the study is unclear and needs to be better described.
It appears that the actual design of the study was as follows: Control group – those with vitamin D serum levels over 30 ng/ml Supplemented group – those with serum vitamin D levels below 30 ng/ml
Answer: We don’t agree with the expert reviewer. Soccer players with the level of the 25(OH)D more than 30 ng/ml were excluded from the research after the first blood test.We studied the effect of the oral vitamin D supplements only in groups with insufficiency (21-29 ng/ml) and deficiency (below 20 ng/ml), because we wanted to study the effect of correction by equal doses depending on the baseline level of 25(OH)D.
If you had supplemented the insufficient and deficient groups with different doses of vitamin D, they would have represented different groups.
Answer: We agree with the expert reviewer, but we wanted to evaluate the impact of the equal doses of the oral vitamin D supplements on the efficiency of the correction depending on the baseline level of 25(OH)D.In our next works (winter 2020) we will evaluate the effect of different doses of the vitamin D.
It also appears that the Authors have taken blood samples from 131 participants and then excluded over 50% of the participants based upon their vitamin D sufficiency levels. This has not been clearly stated in the methodology or in the exclusion criteria.
Can you please explain if this is actually what happened?
Answer: We agree with the expert reviewer. We didn’t study the dynamic of the changes of serum concentration of 25(OH)D in group with normal level of vitamin D, because the main purpose of the study was to find out the prevalence of deficiency and insufficiency of vitamin D and to evaluate the correction’s efficiency of this condition. For this we didn’t need to estimate serum concentration of 25(OH)D among athletes with normal concentration.
Please explain why you selectively measured muscle and body fat measurements for insufficient and deficient players only. Please explain why you chose to implement a five-day break in the middle of the supplementation period. Please explain whether all participants had blood taken at every time point, and if not, explain why.
Answer: we examined body composition among athletes with high, insufficient and deficient level of 25(OH)D. We can paste these data in the text in the results. We made a five-day break in the supplementation period after the 30 days of correction, because we were going to make the second blood test, so we didn’t want to distort the real serum concentration of the 25(OH)D, by the intake of the vitamin D.
Lines 145 – 193: The authors have selectively reported the results of those participants that received a supplement and have not presented the results of those participants who presented with normal levels at baseline with the exception of reporting the baseline vitamin D status.
It is unclear whether those participants who were deemed to be sufficient were included in the study beyond baseline.
It would have been informative to see what the progression of vitamin D levels were for all participants over the course of the intervention.
You should include ALL results in your paper and provide explanation in the Discussion section.
Answer: We agree with the expert reviewer. All the soccer players with the level of the 25(OH)D 30 ng/ml and more were excluded from the research after the first blood test, because the main purpose of the study was to evaluate the correction’s efficiency of insufficiency and deficiency of vitamin D. In our future researches we will separately give the information about changes in the serum concentration of 25(OH)D among professional soccer players in winter, spring, summer and autumn.
Figure 1 – please change to patterned areas not color as the sections of the pie chart cannot be distinguished if printed in black and white.
Answer: We agree with the expert reviewer and changed as requested.
Lines 194 to 252:
The discussion has not compared the study’s results well to other literature.
Please provide discussion to explain what mechanisms may have caused the results in the second month to be lower than the first month.
Please provide discussion regarding limitations to the study.
Answer: We agree with the expert reviewer. We can’t explain exactly the decrease of the vitamin D concentrations in the second month. Maybe the reason is the accumulated effect of the short daylight hours during December-February period in Moscow. During these months, daylight hours continue for 7–9 hours, and most of the study participants spent this time indoors during training sessions. Thus, UV exposure was minimal.
Line 200: please explain what “other biochemical agents” you are referring to.
We agree with the expert reviewer and changed as requested.
Line 203: references for “other studies”
We agree with the expert reviewer and changed as requested.
Line 226: “According to the available data” – please provide a reference and explain why you used a cut-off point of 30 ng/ml if the concentration should exceed 40 ng/ml.
Answer: We agree with the expert reviewer. More often the «normal» level is considered to be from 30 ng/ml. The level 40 ng/ml and higher could be «normal» level for the function of the muscle tissues, however these data also could be disputable. And in this research, we didn’t want to show and evaluate the connection between the serum concentration of 25(OH)D and muscle function. So, we decided to take the level higher than 30 ng/ml as normal.
Conclusion
I am unsure how the findings support the conclusion. Please clarify.
We agree with the expert reviewer and changed as requested.
Reviewer 2 Report
Vitamin D insufficiency and deficiency are highly prevalent in adult football players and this situation may be worse in some regions such as Canada or Russia. If soccer is not the primarily favored sport in Canada, it is often played a lot among Russian youth. Vitamin D insufficiency and deficiency can exceed 80% in some regions. The manuscript is interesting but can be improved making analysis with historical Russian data, a comparative analysis with Scotland and data may be found on the web or requested, and identifying the opportunity to search SNPs. It is important to make a paragraph discussing the role of SNPs in vitamin D deficiency and insufficiency. Moreover, vitamin D metabolism is also important. How was the renal and liver function in the patients and can you please discuss the setting of alcoholic liver disease and vitamin D derailment? There is no mention of the drinking habit. Please add and comment on the drinking habit. There is frank evidence of NAFLD/AFLD and Vitamin D deficiency. Please consider to cite these references:
Testino G, Leone S, Fagoonee S. Alcoholic liver disease and vitamin D deficiency. Minerva Med. 2018 Oct;109(5):341-343. doi: 10.23736/S0026-4806.18.05732-4. Epub 2018 Jul 2. PubMed PMID: 29963832. Sangouni AA, Ghavamzadeh S, Jamalzehi A. A narrative review on effects of vitamin D on main risk factors and severity of Non-Alcoholic Fatty Liver Disease. Diabetes Metab Syndr. 2019 May - Jun;13(3):2260-2265. doi: 10.1016/j.dsx.2019.05.013. Epub 2019 May 22. Review. PubMed PMID: 31235166. Barrea L, Muscogiuri G, Annunziata G, Laudisio D, de Alteriis G, Tenore GC, Colao A, Savastano S. A New Light on Vitamin D in Obesity: A Novel Association with Trimethylamine-N-Oxide (TMAO). Nutrients. 2019 Jun 10;11(6). pii: E1310. doi: 10.3390/nu11061310. PubMed PMID: 31185686; PubMed Central PMCID: PMC6627576. Corfield A, Meyer P, Kassam S, Mikuz G, Sergi C. SNPs: At the origins of the databases of an innovative biotechnology tool. Front Biosci (Schol Ed). 2010 Jan 1;2:1-4. Review. PubMed PMID: 20036923.Author Response
Vitamin D insufficiency and deficiency are highly prevalent in adult football players and this situation may be worse in some regions such as Canada or Russia. If soccer is not the primarily favored sport in Canada, it is often played a lot among Russian youth. Vitamin D insufficiency and deficiency can exceed 80% in some regions. The manuscript is interesting but can be improved making analysis with historical Russian data, a comparative analysis with Scotland and data may be found on the web or requested, and identifying the opportunity to search SNPs. It is important to make a paragraph discussing the role of SNPs in vitamin D deficiency and insufficiency. Moreover, vitamin D metabolism is also important. How was the renal and liver function in the patients and can you please discuss the setting of alcoholic liver disease and vitamin D derailment? There is no mention of the drinking habit. Please add and comment on the drinking habit. There is frank evidence of NAFLD/AFLD and Vitamin D deficiency. Please consider to cite these references:
Testino G, Leone S, Fagoonee S. Alcoholic liver disease and vitamin D deficiency. Minerva Med. 2018 Oct;109(5):341-343. doi: 10.23736/S0026-4806.18.05732-4. Epub 2018 Jul 2. PubMed PMID: 29963832. Sangouni AA, Ghavamzadeh S, Jamalzehi A. A narrative review on effects of vitamin D on main risk factors and severity of Non-Alcoholic Fatty Liver Disease. Diabetes Metab Syndr. 2019 May - Jun;13(3):2260-2265. doi: 10.1016/j.dsx.2019.05.013. Epub 2019 May 22. Review. PubMed PMID: 31235166. Barrea L, Muscogiuri G, Annunziata G, Laudisio D, de Alteriis G, Tenore GC, Colao A, Savastano S. A New Light on Vitamin D in Obesity: A Novel Association with Trimethylamine-N-Oxide (TMAO). Nutrients. 2019 Jun 10;11(6). pii: E1310. doi: 10.3390/nu11061310. PubMed PMID: 31185686; PubMed Central PMCID: PMC6627576. Corfield A, Meyer P, Kassam S, Mikuz G, Sergi C. SNPs: At the origins of the databases of an innovative biotechnology tool. Front Biosci (Schol Ed). 2010 Jan 1;2:1-4. Review. PubMed PMID: 20036923.
Answer: We agree with the expert reviewer. The topic of alcoholic liver disease (ALD) and nonalcoholic fatty liver disease (NAFLD) and serum concentration of vitamin D is very interesting, but participants of our research are young soccer players, who don’t abuse alcohol and they don’t have NAFLD/AFLD. So, we decided not to expand the introduction part for discussion this topic, but in our future works with elderly soccer players, we will develop this theme for sure.
Reviewer 3 Report
I appreciate the opportunity to review this manuscript submitted to Nutrients. The authors sought to evaluate vitamin D status as measured via 25(OH)D in young Russian male soccer players, and the extent to which supplementing with 5,000 IUs per day of cholecalciferol would improve 25(OH)D levels. I have outlined my primary concerns with the manuscript below, followed by more specific comments and suggestions.
GENERAL CONCERNS/COMMENTS
--The authors are inconsistent with the term they use to describe the sport of their athletes (football vs. soccer). Pick one and stick with it.
--The authors give the impression that there is a clear threshold for when 25(OH)D begins to impact muscle function. However, it is not entirely clear what the thresholds are for the muscle development and performance effects of 25(OH)D. For example, one of the references used (#7 Shuler 2012) reports that optimal musculoskeletal benefits occur at 25(OH)D levels above the current definition of sufficiency (> 30 ng/mL) with no reported sports benefits above 50 ng/mL. Another reference (#16, Butscheidt 2017), however, reports that levels above 50 ng/ml might be required for athletes to achieve maximal physical performance. Clearly, there is uncertainty on this issue in the literature, and the authors need to do a better job of conveying that.
----The introduction section it too lengthy and unfocused. I count 9 paragraphs, and this could easily be reduced to 6 or 7 paragraphs.
ABSTRACT
--pg 1, line 31: Define for the reader how low vitamin D levels were classified. Also, I’m assuming the authors used 25(OH)D to assess vitamin D status. Although they mention 25(OH)D at the end of the abstract, they should clarify this earlier in the abstract.
--pg 1, lines 31-32: Do the authors mean to say 5,000 IU per day? Specify the frequency of supplementation.
--pg 1, lines 32-33: Report the average absolute increase and not just the % increase.
--pg 1, line 36: What is meant by “both groups”? I don’t see anywhere in the abstract where the authors define specific groups.
--pg 1, line 38: I’m not sure everyone would agree that 5,000 IU/day is a small dosage. Instead of using a vague descriptor, just restate the dose.
INTRODUCTION
--pg 2, line 47: “Muscle tone” is not a scientific phrase. Please reword.
--pg 2, line 53: When the authors talk about adequate vitamin D levels, they should specify that they are basing their discussion on 25(OH)D.
--pg 2, lines 55-56: The notion that vitamin D deficiency is a pandemic is controversial. While many observational studies link low vitamin D levels to various health issues, data from RCTs are less convincing. To me, this statement is hyperbolic and largely unnecessarily.
--pg 2, lines 57-58: The authors should specify that UVB ray exposure is most important. Also, these two sentences need supportive references.
--pg 2, lines 67-68: It is not entirely clear whether 40 and 50 ng/mL are the actual thresholds for the muscle development and performance effects. They are hypothesized suggestions, so the authors should be more cautious with their statements on this subject.
--pg 2, lines 69-78: This section of text is a somewhat disorganized and scattered. The authors should rework it for clarity.
--pg 2, line 74: What is the specific evidence for the following statement: “Exercises increase the risk of developing of vitamin D deficiency even if you take supplements”?
--pg 2, lines 93-94: When I look at the full text of reference 27, I don’t see anything that supports this statement.
METHODS
--pg 3 lines 116-117: When the authors report that players who received other supplements were excluded, can they be more specific as to why this was done, and which supplements triggered exclusion? Also, were players who chose to supplement on their own also excluded?
--pg 3, line 117: Define the acronym ARVI.
--pg 3, lines 143-144: This section needs more detail. The authors should specify when and how each of these tests was used.
RESULTS
--pg 4, line149: In the methods section, the authors state that, “Values of 30-60 ng/ml were considered normal.” However, here they are also including athletes above 60 as normal. The authors should be consistent as to whether they consider above 60 ng/ml as normal.
--pg 4, Table 1: The authors should somehow indicate which statistical tests were used for these comparisons.
--pg 5, lines 174-189: There needs to be some additional clarity as to the sample sizes for each group during the follow-up period. Previously on pg 4, the authors stated that 7 and 3 of the players were unavailable for the second and third sampling. I suggest describing in the figure or in the figure legend the sample sizes for each time point.
--pg 5, lines 191-193: The justification for the statistics should be in the Methods section, not here.
DISCUSSION
--pg 5, lines 195-200: These two paragraphs seem out of place and are perhaps better placed in the Methods or Introduction sections.
--pg 6, lines 221-222: Perhaps I am missing it in the Holick paper, but I don’t see any discussion in that paper about how exercise volume impacts vitamin D status.
--pg 6, line 239: You can delete the word “any”.
--pg 6, line 242: Using the phrase “vitamin level” is not specific enough. Report the actual biomarker name (I’m assuming it is 25(OH)D).
CONCLUSION
--pgs 7-8, lines 254-257: The concluding statements are vague and don’t directly reflect what the study can reasonably say about vitamin D supplementation. Instead, the concluding statement should more closely resemble what the authors write in the abstract (i.e., “The prevalence of vitamin D deficiency was high in young Russian football players. Furthermore, it was indicated that the daily usage of cholecalciferol in small doses was an effective and well-tolerated treatment for vitamin D insufficiency. No linear dependency between the duration of treatment and increase in vitamin 25(OH)D concentration was observed.”)
Author Response
I appreciate the opportunity to review this manuscript submitted to Nutrients. The authors sought to evaluate vitamin D status as measured via 25(OH)D in young Russian male soccer players, and the extent to which supplementing with 5,000 IUs per day of cholecalciferol would improve 25(OH)D levels. I have outlined my primary concerns with the manuscript below, followed by more specific comments and suggestions.
GENERAL CONCERNS/COMMENTS
--The authors are inconsistent with the term they use to describe the sport of their athletes (football vs. soccer). Pick one and stick with it.
Answer: We agree with the expert reviewer and changed as requested.
--The authors give the impression that there is a clear threshold for when 25(OH)D begins to impact muscle function. However, it is not entirely clear what the thresholds are for the muscle development and performance effects of 25(OH)D. For example, one of the references used (#7 Shuler 2012) reports that optimal musculoskeletal benefits occur at 25(OH)D levels above the current definition of sufficiency (> 30 ng/mL) with no reported sports benefits above 50 ng/mL. Another reference (#16, Butscheidt 2017), however, reports that levels above 50 ng/ml might be required for athletes to achieve maximal physical performance. Clearly, there is uncertainty on this issue in the literature, and the authors need to do a better job of conveying that.
Answer: We agree with the expert reviewer and changed as requested.
----The introduction section it too lengthy and unfocused. I count 9 paragraphs, and this could easily be reduced to 6 or 7 paragraphs.
Answer: We agree with the expert reviewer. But, on the contrary, others asked us to add some more information. We will do our best to reorganize the introduction and make it more specific.
ABSTRACT
--pg 1, line 31: Define for the reader how low vitamin D levels were classified. Also, I’m assuming the authors used 25(OH)D to assess vitamin D status. Although they mention 25(OH)D at the end of the abstract, they should clarify this earlier in the abstract.
Answer: We agree with the expert reviewer and changed as requested.
--pg 1, lines 31-32: Do the authors mean to say 5,000 IU per day? Specify the frequency of supplementation.
Answer: We agree with the expert reviewer and changed as requested.
--pg 1, lines 32-33: Report the average absolute increase and not just the % increase.
Answer: We agree with the expert reviewer and changed as requested.
--pg 1, line 36: What is meant by “both groups”? I don’t see anywhere in the abstract where the authors define specific groups.
Answer: We agree with the expert reviewer. We meant group with deficiency and group with insufficiency. We will add it in the text.
--pg 1, line 38: I’m not sure everyone would agree that 5,000 IU/day is a small dosage. Instead of using a vague descriptor, just restate the dose.
Answer: We agree with the expert reviewer and changed as requested.
INTRODUCTION
--pg 2, line 47: “Muscle tone” is not a scientific phrase. Please reword.
Answer: We agree with the expert reviewer and changed as requested.
--pg 2, line 53: When the authors talk about adequate vitamin D levels, they should specify that they are basing their discussion on 25(OH)D.
Answer: We agree with the expert reviewer and changed as requested.
--pg 2, lines 55-56: The notion that vitamin D deficiency is a pandemic is controversial. While many observational studies link low vitamin D levels to various health issues, data from RCTs are less convincing. To me, this statement is hyperbolic and largely unnecessarily.
Answer: We agree with the expert reviewer and changed as requested.
--pg 2, lines 57-58: The authors should specify that UVB ray exposure is most important. Also, these two sentences need supportive references.
Answer: We agree with the expert reviewer and changed as requested.
--pg 2, lines 67-68: It is not entirely clear whether 40 and 50 ng/mL are the actual thresholds for the muscle development and performance effects. They are hypothesized suggestions, so the authors should be more cautious with their statements on this subject.
Answer: We agree with the expert reviewer and changed as requested.
--pg 2, lines 69-78: This section of text is a somewhat disorganized and scattered. The authors should rework it for clarity.
Answer: We agree with the expert reviewer and changed as requested.
--pg 2, line 74: What is the specific evidence for the following statement: “Exercises increase the risk of developing of vitamin D deficiency even if you take supplements”?
Answer: It was our assumption, because during hard physical work, training, human body expend more energetic resources, vitamins than during daily activities. Thus, the athlete’s organism needs more vitamins, including vitamin D.
--pg 2, lines 93-94: When I look at the full text of reference 27, I don’t see anything that supports this statement.
Answer: «It has been estimated that the body uses daily on average 3,000–5,000 IU of cholecalciferol» We will change «athlete’s» on «human».
METHODS
--pg 3 lines 116-117: When the authors report that players who received other supplements were excluded, can they be more specific as to why this was done, and which supplements triggered exclusion? Also, were players who chose to supplement on their own also excluded?
Answer: We agree with the expert reviewer. We excluded the intake of all other supplements in order to exclude their influence on running speed and muscle strength in the research that was carried out at same time with this group of athletes of this study. The most frequent supplements among young soccer players are BCAA, protein and carbohydrate gels. All that they take in is given only by doctors from the academy. Self-supplementation is prohibited.
--pg 3, line 117: Define the acronym ARVI.
Answer: It is a typo. We will fix it. We meant acute respiratory viral infections.
--pg 3, lines 143-144: This section needs more detail. The authors should specify when and how each of these tests was used.
Answer: We agree with the expert reviewer and changed as requested.
RESULTS
--pg 4, line149: In the methods section, the authors state that, “Values of 30-60 ng/ml were considered normal.” However, here they are also including athletes above 60 as normal. The authors should be consistent as to whether they consider above 60 ng/ml as normal.
Answer: We agree with the expert reviewer and changed as requested.
--pg 4, Table 1: The authors should somehow indicate which statistical tests were used for these comparisons.
Answer: We agree with the expert reviewer and changed as requested.
--pg 5, lines 174-189: There needs to be some additional clarity as to the sample sizes for each group during the follow-up period. Previously on pg 4, the authors stated that 7 and 3 of the players were unavailable for the second and third sampling. I suggest describing in the figure or in the figure legend the sample sizes for each time point.
Answer: We agree with the expert reviewer and changed as requested.
--pg 5, lines 191-193: The justification for the statistics should be in the Methods section, not here.
Answer: We agree with the expert reviewer and changed as requested.
DISCUSSION
--pg 5, lines 195-200: These two paragraphs seem out of place and are perhaps better placed in the Methods or Introduction sections.
Answer: We agree with the expert reviewer and changed as requested.
--pg 6, lines 221-222: Perhaps I am missing it in the Holick paper, but I don’t see any discussion in that paper about how exercise volume impacts vitamin D status.
Answer: We agree with the expert reviewer and changed as requested.
--pg 6, line 239: You can delete the word “any”.
Answer: We agree with the expert reviewer and changed as requested.
--pg 6, line 242: Using the phrase “vitamin level” is not specific enough. Report the actual biomarker name (I’m assuming it is 25(OH)D).
Answer: We agree with the expert reviewer and changed as requested.
CONCLUSION
--pgs 7-8, lines 254-257: The concluding statements are vague and don’t directly reflect what the study can reasonably say about vitamin D supplementation. Instead, the concluding statement should more closely resemble what the authors write in the abstract (i.e., “The prevalence of vitamin D deficiency was high in young Russian football players. Furthermore, it was indicated that the daily usage of cholecalciferol in small doses was an effective and well-tolerated treatment for vitamin D insufficiency. No linear dependency between the duration of treatment and increase in vitamin 25(OH)D concentration was observed.”)
Answer: We agree with the expert reviewer and changed as requested.
Reviewer 4 Report
The authors presented an interesting work, however some major changes are needed.
ABSTRACT
What authors mean for "low", be accurate. Insufficient or deficient?
Again with "reference range", be specific.
INTRODUCTION
Nice paragraph. I would suggest to better reorganize the rationale of this paragraph (a clear order should be: Vitamin D definition, Vitamin function, Vitamin d insuff/defic prevalence [for regiona, latitude and so on..,], and then the role of supplementation for athletes. Please take in mind that the correct chronobiologic approach to the problem could provide important solution, see and cite for example: Vitale et al., 2018, Chronobiol Int (with skiers). Authors could also describe the role of Vitamin D for sport performance.
Need to be more specific for the study aims, mostly with reference to the supplementation.
An hypothesis in missing.
METHODS
Reorganize this paragraph in subsections: subjects, blood sampling, statistical analysis.
Were the subjects real athletes? Please give further details on weekly training volume and competition level.
A flow chart (with drop-out subjects too) may help to understand the timing of evaluations and study design.
Statistical analysis is not good enough, further details are needed. Were t-test and Mann-Whytney or Wilcoxon used for...? How do authors compare improvement from PRE to +30 and + 60? I Would suggest a 1 way ANOVA or 2-way ANOVA if 2 diferents groups are compared (and insert in fugure 2 p-values)
RESULTS
This is a suggestion: Figures should be done again with different software.
Mean values in Figure 2 are not correct for the combined group.
Insert in figure 2 standard deviations. Also p-values are missing
Probably a Table could help to show details of mean and SD of different evaluations (PRE, + 30 + 60).
Need to reorganize figure 2 and probably insert the results of 2-way ANOVA (see comments above).
DISCUSSION AND CONCLUSION
This paragraph should be rewritten after new results.
Author Response
The authors presented an interesting work; however, some major changes are needed.
ABSTRACT
What authors mean for "low", be accurate. Insufficient or deficient?
Answer: We meant both of these conditions.
Again with "reference range", be specific.
Answer: We agree with the expert reviewer and changed as requested.
INTRODUCTION
Nice paragraph. I would suggest to better reorganize the rationale of this paragraph (a clear order should be: Vitamin D definition, Vitamin function, Vitamin d insuff/defic prevalence [for regiona, latitude and so on..,], and then the role of supplementation for athletes. Please take in mind that the correct chronobiologic approach to the problem could provide important solution, see and cite for example: Vitale et al., 2018, Chronobiol Int (with skiers). Authors could also describe the role of Vitamin D for sport performance.
Answer: We agree with the expert reviewer and changed as requested.
Need to be more specific for the study aims, mostly with reference to the supplementation.
Answer: We agree with the expert reviewer and changed as requested.
An hypothesis in missing.
Answer: We agree with the expert reviewer and changed as requested.
METHODS
Reorganize this paragraph in subsections: subjects, blood sampling, statistical analysis.
Were the subjects real athletes? Please give further details on weekly training volume and competition level.
Answer: We agree with the expert reviewer and changed as requested.
A flow chart (with drop-out subjects too) may help to understand the timing of evaluations and study design.
Answer: We agree with the expert reviewer and changed as requested.
Statistical analysis is not good enough, further details are needed. Were t-test and Mann-Whytney or Wilcoxon used for...? How do authors compare improvement from PRE to +30 and + 60? I Would suggest a 1 way ANOVA or 2-way ANOVA if 2 diferents groups are compared (and insert in fugure 2 p-values)
Answer: We agree with the expert reviewer and changed as requested.
RESULTS
This is a suggestion: Figures should be done again with different software.
Answer: We agree with the expert reviewer and changed as requested.
Mean values in Figure 2 are not correct for the combined group.???
Answer: We agree with the expert reviewer and changed as requested.
Insert in figure 2 standard deviations. Also p-values are missing
Answer: We agree with the expert reviewer and changed as requested.
Probably a Table could help to show details of mean and SD of different evaluations (PRE, + 30 + 60).
Answer: We agree with the expert reviewer and changed as requested.
Need to reorganize figure 2 and probably insert the results of 2-way ANOVA (see comments above).
Answer: We agree with the expert reviewer and changed as requested.
DISCUSSION AND CONCLUSION
This paragraph should be rewritten after new results.
Answer: We agree with the expert reviewer and changed as requested.
Round 2
Reviewer 1 Report
You have not addressed my concerns regarding the design of your study or properly explained the exclusion of the 'adequate' group.
Author Response
You have not addressed my concerns regarding the design of your study or properly explained the exclusion of the 'adequate' group.
Answer: We agree with the expert reviewer. We ascribed it to the limitations of our study and will consider it in our future works for sure. The main purpose of this research was to study the prevalence of vitamin D deficiency and insufficiency among young football players. Also, we wanted to evaluate the effectiveness of correction of the low content of 25(OH)D with daily supplementation of oral cholecalciferol. That is why we decided not to re-evaluate the serum concentration of 25(ОН)D in the «adequate» group and they were excluded from the further part of the study.
Reviewer 4 Report
I would like to compliment authors for their work of revision. Few comments now:
As suggested before, the authors did not cite the following papers: https://www.ncbi.nlm.nih.gov/pubmed/29231753. Categories of figure 1 are too big, please modify. Stress again practical applications.Author Response
I would like to compliment authors for their work of revision. Few comments now:
As suggested before, the authors did not cite the following papers: https://www.ncbi.nlm.nih.gov/pubmed/29231753.
Categories of figure 1 are too big, please modify.
Stress again practical applications.
Answer: We agree with the expert reviewer and changed a suggested.